# Sarm1 knockout protects against early but not late axonal degeneration in experimental allergic encephalomyelitis

**Kenneth Viar, Daniel Njoku, Julie Secor McVoy, Unsong Oh** *

Department of Neurology, Virginia Commonwealth University School of Medicine, Richmond, Virginia, United States of America

* uoh@vcu.edu

## Abstract

Programmed axonal degeneration, also known as Wallerian degeneration, occurs in immune-mediated central nervous system (CNS) inflammatory disorders such as multiple sclerosis and the animal model experimental allergic encephalomyelitis (EAE). Sterile alpha and TIR domain containing protein 1 (SARM1) functions to promote programmed axonal degeneration. To test the hypothesis that loss of SARM1 will reduce axonal degeneration in immune-mediated CNS inflammatory disorders, the course and pathology of EAE was compared in Sarm1 knockout mice and wild type littermates. The clinical course of EAE was similar in Sarm1 knockout and wild type. Analysis of EAE in mice expressing neuronal yellow fluorescent protein (YFP) showed significantly less axonal degeneration in Sarm1 knockout mice compared to wild type littermates at 14 days post-induction of EAE. At 21 days post-induction, however, difference in axonal degeneration was not significant. At 42 days post-induction, Sarm1 knockout mice were indistinguishable from wild type with respect to markers of axonal injury, and were similar with respect to axonal density in the lumbar cords. There was no significant change in peripheral immune activation or CNS inflammatory cell infiltration associated with EAE in Sarm1 knockout mice. In conclusion, Sarm1 deletion delayed axonal degeneration early in the course of CNS inflammation, but did not confer long-term protection from axonal degeneration in an animal model of immune-mediated CNS inflammation.

**Data Availability Statement:** Data are held in Open Science Framework repository: osf.io/5mzr6

**Funding:** This study was supported by Virginia Commonwealth University Excellence Fund

## Introduction

Axonal degeneration is a major determinant of disability accumulation in chronic CNS inflammatory disorders such as multiple sclerosis [1,2]. Programmed axonal degeneration, also known as Wallerian degeneration, is known to occur in multiple sclerosis and in the animal model EAE [3,4]. However, whether or not targeting programmed axonal degeneration confers protection from long-term axonal loss in immune-mediated CNS inflammatory disorders such as multiple sclerosis and EAE remains unconfirmed.

(internal funding to U.O.). There was no additional external funding received for this study.

**Competing interests:** The authors have declared that no competing interests exist.

Proteins involved in nicotinamide adenine dinucleotide (NAD) metabolism such as SARM1 and nicotinamide mononucleotide adenylyltransferase (NMNAT) were identified as key opposing factors that control programmed axonal degeneration [5,6]. SARM1 activation results in derepression of its NADase activity, which resides in its Toll/interleukin-1 (TIR) domain, leading to intra-axonal NAD depletion, local energetics failure and axonal degeneration [7]. SARM1-dependent neurodegeneration has been implicated in a number of different models of neuronal injury based on the finding that *Sarm1* deletion was neuroprotective in models of traumatic, toxic and metabolic nervous system injury [8–10]. Thus prior research suggested that SARM1-dependent program of axonal degeneration might be a common mechanism contributing to neurodegeneration in a variety of contexts.

In addition to the function of neuronal SARM1 in programmed axonal degeneration, SARM1 expression in immune cells has the potential to influence immune response through its TIR domain, which can modulate Toll-like receptor signaling [11]. Prior reports suggested that the effect of SARM1 inactivation on the immune response depends on disease model or species studied. In human peripheral blood leukocytes, SARM1 functions to inhibit immune responses [11–13]. In animal models of CNS infections, however, *Sarm1* deletion led to reduced CNS inflammation, suggesting that SARM1 functions to augment CNS anti-viral response [14,15]. The role of SARM1 in the immune response associated with immune-mediated "sterile" CNS inflammation such as EAE is unknown.

To test the hypothesis that *Sarm1* deletion would reduce axonal degeneration in CNS inflammatory disorders, the course and pathology of EAE in *Sarm1* knockout mice and wild type littermates were compared. Clinical course and markers of axonal degeneration were assessed. Immune activation and inflammatory infiltrates were also assessed to exclude the possibility that changes in the neuroimmunology of EAE in *Sarm1* knockout mice might influence the outcome. We found that the incidence and clinical course of EAE were similar in *Sarm1* knockout mice and wild type littermates. *Sarm1* knockout mice showed substantially less axonal injury early in the course of EAE. However, the extent of axonal degeneration did not differ substantially later in the course of EAE.

## Materials and methods

### Reagents

All reagents were from Thermo Fisher Scientific unless otherwise specified.

### Animals

*Sarm1* knockout mice (B6.129X1-Sarm1^tm1Aidi/J) were obtained from Jackson Laboratory (Bar Harbor, ME) and maintained on a C57BL/6J background. Heterozygous mating produced homozygous *Sarm1* knockout (i.e. *Sarm1*-/-), heterozygotes (i.e. Sarm1+/-) and wild type (i.e. *Sarm1*+/+) littermates. A colony of *Sarm1* knockout mice and wild type littermates expressing neuronal YFP was generated by crossing *Sarm1* knockout mice with Thy1-YFP-H transgenic mice (B6.Cg-Tg(Thy1-YFP)HJrs/J, Jackson Laboratory). *Sarm1*-/-YFP+ mice and *Sarm1*+/+YFP+ littermates were generated from *Sarm1*+/-YFP+ x *Sarm1*+/- mating pairs. Mice were genotyped using the following PCR primers. *Sarm1* knockout forward and reverse primers were `CTT GGG TGG AGA GGC TAT TC` and `AGG TGA GAT GAC AGG AGA TC`, respectively. Wild type forward and reverse primers were `GGG AGA GCC TTC CTC ATA CC` and `TAA GGA TGA ACA GGG CCA AG`, respectively. YFP transgene expression was detected by the presence of neuronal YFP in ear punch samples under fluorescence microscopy. Animals were housed in littermate groups regardless of genotype, on a 12 h light-dark cycle and fed ad libitum. All animal procedures were performed in accordance with the Virginia

Commonwealth University Animal Care and Use Program's regulations under an approved protocol (protocol number: AD10000395).

## Neuronal cultures, axotomy and oxidative stress

Cortex from E15.5 embryos of *Sarm1* knockout and C57BL/6J mice were dissected then enzymatically dissociated in neurobasal medium containing trypsin (2.5 mg/ml, Sigma Aldrich) and DNase I (15 µg/ml, Sigma Aldrich) for 30 minutes at 37°C, washed in neurobasal medium, then triturated using fire-polished Pasteur pipettes. Cell suspensions were passed through a 70 µm strainer to remove debris. Cells were plated on poly-D-lysine (Sigma Aldrich) coated plates or glass-bottomed dish and cultured in neuronal culture media (neurobasal medium supplemented with B27, GlutaMax (0.5 mM) and penicillin/streptomycin/amphotericin B). Neuronal culture media were changed by one-half volume exchange every 3 days. Cytarabine (1 µM final, Sigma Aldrich) was added from day in vitro (DIV) 3 to 6 to inhibit glial proliferation. Axotomy was performed manually on DIV 12 to 14 neurons cultured in 35 mm glass-bottomed dish using the cutting edge of a 27-gauge sterile needle under visual guidance through a low magnification inverted light microscope. Neurons (DIV 10 to 12) were subjected to oxidative stress by 30-minute exposure to hydrogen peroxide ($H_2O_2$). First, neuronal culture media was replaced by 2 x volume exchange with plain neurobasal media. $H_2O_2$ in plain neurobasal media was added to wells by ½ media exchange to indicated final concentrations. After 30 minutes of exposure at 37°C, media was replaced by 2 x volume exchange to neuronal culture media. Cells were analyzed 24 hours later. Viability of cultured neurons was assessed by fluorescent dye exclusion (Ready Probes Cell Viability Imaging Kit Blue/Green) and expressed as percent of control (0 µM $H_2O_2$).

## Neuronal immunocytochemistry

Tau, NeuN and ankyrin G were detected in cultured neurons by immunocytochemistry. Neurons were fixed for 10 minutes with 4% (w/v) paraformaldehyde, then washed 3 times in phosphate buffered saline (PBS). Cells were permeabilized in PBS containing 0.3% Triton-X100 and 10% normal goat serum, then incubated with primary antibodies against Tau (clone EP2456Y, Abcam, RRID: AB_1524475), NeuN antibody (clone A60, Millipore Sigma, RRID: AB_2298772) and ankyrin G (clone N106/36, Millipore Sigma, RRID: AB_2749806) for 1 hour at room temperature. After washing in PBS, cells were incubated with fluorochrome conjugated secondary antibodies for 1 hour at room temperature, then washed in PBS prior to microscopy. Tau immunocytochemistry was imaged on Zeiss LSM710 confocal laser scanning microscope on 40x/1.3 NA oil-immersion objective using a pin hole of 1 Airy disc unit and Nyquist sampling. For NeuN and ankyrin G immunocytochemistry, 3 non-overlapping fields of view were imaged from each well on a FLoid Cell Imaging Station using a 20x/0.45 NA objective.

## EAE induction and clinical scoring

EAE was actively induced in 6- to 10-week old *Sarm1* knockout mice and wild type littermates by subcutaneous injection of 200 µg of myelin oligodendrocyte glycoprotein peptide 35–55 (MOG$_{35-55}$, Anaspec) emulsified in complete Freund's adjuvant (CFA) containing 500 µg M. Tuberculosis H37 RA (Difco) and intraperitoneal injection of 300 ng pertussis toxin (List Biological Laboratories), followed by a second intraperitoneal injection of 300 ng pertussis toxin 2 days later. Female and male mice were induced. Animals that died prior to onset of clinical EAE were included in the mortality analysis, but excluded from analysis of EAE clinical scores. Mice were scored daily as follows: 0 –no overt signs of disease; 1 –limp tail or loss of righting

reflex but not both; 2 –limp tail and loss of righting reflex; 3 –partial hind limb paralysis; 4 –complete hind limb paralysis; 5 –moribund state or death. Clinical score rater was blinded to genotype. Primary clinical outcome was the mean total or cumulative clinical score defined as the sum of daily clinical scores from induction until 42 days post-induction [16].

## Perfusion fixation and tissue processing

Following humane killing, animals underwent transcardiac perfusion with up to 50 ml of normal saline followed by perfusion with 100 ml of 4% (w/v) paraformaldehyde in PBS using a rate controlled pump. Following fixation, the entire CNS tissue was dissected out then cryoprotected in 30% sucrose in PBS for over 48 h. Spinal cords were cut into three 1 cm length sections measured from the cervicomedullary junction caudally under a dissecting microscope to approximate cervical, thoracic and lumbar spinal cord segments. Lumbar cord was further cut into two 0.5 cm segments—rostral half for transverse sections and caudal half for coronal sections. Tissue were then cryopreserved at -80˚C in Optimal Cutting Temperature compound. Serial 20 μm sections through the entire lumbar cord in a ventral to dorsal direction (coronal sections) was obtained for each animal, and every 5th slide (~300 μm interval) was processed for YFP+ axon analysis or immunohistochemistry. Alternatively, transverse sections of lumbar cords at ~300 μm intervals were processed for SMI-31 immunohistochemistry.

## Analysis of YFP+ axons

Coronal sections from the lumbar cords of EAE-induced *Sarm1*-/-YFP+ and *Sarm1*+/+YFP+ mice were rinsed 3 times in Tris buffered saline (TBS), them mounted with anti-fade mounting media (Vectashield, Vector Laboratories) and coverslips. Sections were imaged on a FLoid Cell Imaging Station using a 20x/0.45 NA objective. Six non-overlapping fields of view were acquired from each section. Slides and image files were coded to blind research personnel to genotype. Image analysis was performed using ImageJ by drawing 3 equidistant vertical lines over each image. Intact and fragmented axons that crossed each line was manually counted then averaged for each image. About 24 images were available and analyzed for each animal. Results were expressed as fragmented and total YFP+ axons per field of view per animal.

## Immunohistochemistry

Animals were humanely killed at 42 days post-induction for immunohistochemical analysis of axonal degeneration. Tissue were processed as described above. Antigen-retrieval was performed for all samples by incubation of tissue sections in a citric acid buffer for 10 min at 45˚C in a temperature-controlled microwave (BioWave Pro, Pelco), followed by 3 rinses in TBS. For APP immunohistochemistry, nonspecific antibody binding was blocked by 30 min incubation in a TBS blocking solution containing 4% cold water fish skin gelatin and 0.3% Triton-X 100. Sections were then incubated overnight at 4˚C with anti-APP antibody (RRID: AB_2533275). After several washes in TBS, sections were incubated with appropriate secondary antibodies for 90 minutes at room temperature. After several washes, slides were mounted with anti-fade mounting media and coverslips for fluorescence microscopy. For SMI-31 and SMI-32 immunohistochemistry, sections were post-fixed in pre-chilled methanol at -20˚C for 10 min, followed by 3 rinses in TBS prior to antigen-retrieval. Following methanol post-fixation and antigen retrieval, Mouse on Mouse kit (Vector Laboratories) was used according to manufacturer's instructions to block non-specific binding. Sections were incubated overnight at 4˚C with SMI-31 antibody (BioLegend, RRID: AB_2564642) or SMI-32 antibody (BioLegend, RRID: AB_2564641) for immunodetection of phosphorylated and non-phosphorylated neurofilament H, respectively. After several washes in TBS, biotinylated anti-mouse IgG followed by

fluorescein conjugated Avidin D (Vector Laboratories) or Texas Red conjugated Avidin D (Vector Laboratories) were applied for SMI-31 and SMI-32 immunohistochemistry, respectively, according to manufacturer's instructions. After several washes, slides were mounted with anti-fade mounting media and coverslips. DAPI nuclear stain was applied to sections prior to final washes for all immunohistochemistry samples.

## Image acquisition and analysis

For all image acquisition and analysis, slides and image files were coded to blind research personnel to genotype. For APP immunohistochemistry, images were acquired using a Zeiss AxioImager Z2 microscope. Six non-overlapping field of view images from each lumbar cord section were taken with a 20x/0.8 NA objective. About 24 images were available and analyzed per animal. ImageJ was used to perform a region of interest (ROI) analysis comparing APP+ axon areas between the groups. APP+ signal thresholding at 0.1% was applied to each image. Three non-overlapping ROI (150 μm x 150 μm) were applied to each image and each ROI analyzed for APP+ area using the *Analyze Particles* (size 1 to infinity; circularity 0–1) function of ImageJ. Thus about 72 ROIs were analyzed per animal to derive the mean APP+ area per ROI per animal.

For SMI-32 immunohistochemistry, images were acquired on a FLoid Cell Imaging Station using a 20x/0.45 NA objective. Six non-overlapping fields of view were acquired from each section. About 24 images were available and analyzed per animal. Image analysis was performed using ImageJ. SMI-32 signal thresholding at 3% was applied to each image. Three non-overlapping ROI (200 μm x 200 μm) were superimposed on each image. Each ROI was analyzed for SMI-32+ area using the *Analyze Particles* (size 2 to infinity; circularity 0–1) function of ImageJ. Thus about 72 ROIs were analyzed per animal to derive the mean SMI-32+ area per ROI per animal.

For SMI-31 immunohistochemistry, images were acquired on a Zeiss AxioImager Z2 fluorescence microscope equipped with a motorized stage at 10x/0.45 NA objective. A single montage image of the entire transverse section of a lumbar cord was acquired by using the *slide scanning* function of Neurolucida 360 software (MBF Biosciences). Image analysis was performed using ImageJ. Images were converted to 8-bit images. *Auto Threshold* (Max Entropy) function of ImageJ was applied to each image. ROI consisting of an entire hemicord was outlined manually. SMI-31+ area was analyzed using the *Analyze Particles* (size 0 to infinity; circularity 0–1) function of ImageJ. Data from at least 4 hemicords per animal were averaged to obtain the mean SMI-31+ area per hemicord per animal.

## Fluorescence-activated cell sorting (FACS) analysis

Following humane killing, animals underwent transcardiac perfusion with up to 50 ml of normal saline using a rate controlled pump. Thoracolumbar cords (from T2 to caudal end) were expelled out of the spinal column using hydraulic pressure manually applied through a 19-gauge needle and syringe filled with PBS. Cords were minced using a McIlwaine tissue chopper (Mickle Laboratory Eng. Co., UK), then enzymatically dissociated in RPMI media containing 2.5 mg/ml collagenase D (Roche Diagnostics) and DNase I (20 μg/ml, Sigma Aldrich) for 45 minutes at 37˚C with constant rotation. Cells were passed through a 70 μm strainer, washed in RPMI, resuspended in 30% isotonic Percoll (GE Healthcare) in PBS, then centrifuged at 500 x g for 10 min. Supernatant was removed. Cell pellets were washed and resuspended in RPMI. Cells were aliquoted into tubes and washed in FACS buffer (0.1% sodium azide and 2% fetal calf serum in PBS). Cells were incubated with Fc block (anti-mouse CD16/CD32 antibody, BD Biosciences) for 5 minutes prior to addition of fluorochrome-conjugated antibodies against CD3 (clone 17A2, RRID: AB_395700), CD4 (clone GK1.5, RRID: AB_396633), CD8a (clone 53–6.7, RRID: AB_394570), CD11b (clone M1/70, RRID:

AB_396679), CD11c (clone N418, RRID: AB_469590), CD19 (clone 1D3, RRID: AB_10853189), CD45 (clone 30-F11, RRID: AB_465667), Ly6C (clone HK1.4, RRID: AB_2616730) or Ly6G (clone 1A8-Ly6G, RRID: AB_2573307). Following a 30 min incubation in the dark at 4°C, cells were washed in FACS buffer. Count beads (CountBright, Thermo Fisher) were added to each sample to allow absolute count determination. FACS data was acquired on a flow cytometer (FACS Canto, BD Biosciences). Data were analyzed using FlowJo software (FlowJo, LLC). All analyses were performed on singlet cell-gated populations identified on FSC-H and FSC-A dot plots. CD45+ immune cell subsets were identified as follows: total leukocytes (total CD45+), microglia (CD3-Ly6G-CD11b+CD45int), macrophage (CD3-Ly6G-CD11b+CD45hi), pro-inflammatory monocyte/macrophage (Ly6G-CD11b+Ly6C$^{hi}$), myeloid dendritic cells (Ly6G-CD11b+CD11c+), T cells (CD3+, CD3+CD4+ and CD3+CD8+), B cells (CD3-CD19+) and neutrophils (CD45+Ly6G+).

## Ex vivo antigen (MOG$_{35-55}$)-recall response

Following humane killing, spleens were removed from EAE-induced mice at 14 days post-induction. Single cell suspensions of splenocytes were prepared by pushing the spleen through a 70 μm strainer, then resuspending cells in ammonium-chloride-potassium buffer to lyse erythrocytes. Cells were washed and resuspended in RPMI supplemented with antibiotics and 10% fetal calf serum. Cells were plated in round-bottom 96-well plates at 1 x 10$^6$ per well and stimulated for 72 hours with MOG$_{35-55}$ (20 μg/ml), phytohemagglutinin (2% v/v) or PBS at 37°C in 5% CO$_2$ incubator. Supernatant was collected and stored frozen and used later for detection of cytokines interleukin (IL)-4, IL-17 and interferon-γ (IFN-γ) using enzyme linked immunosorbent assay (ELISA) kits (R&D Systems) according to manufacturer's instructions. Control (PBS-treated) samples showed no greater than 51 pg/ml of IFN-γ in any of the ELISA experiments. Splenocytes were fixed and permeabilized using a fixation/permeabilization kit (eBioscience) and used for intracellular antigen staining (Ki67 or FOXP3).

For intracellular staining of FOXP3 and Ki67, fixed and permeabilized splenocytes obtained from *ex vivo* antigen-recall response assay were incubated with normal goat serum (2% in FACS buffer) to block non-specific binding. Antibodies against the following antigens were added for 30 minutes in the dark: CD4 (clone RM4-5, RRID: AB_464896), CD3 (clone 17A2, RRID: AB_395700), CD25 (clone 7D4, RRID: AB_11149306), FOXP3 (clone FJK-16s, RRID: AB_465935) or Ki67 (clone B56, RRID: AB_10611874). Cells were then washed in permeabilization buffer (eBioscience) then resuspended in FACS buffer prior to FACS analysis.

## Statistical analysis

Estimation statistics (www.estimationstats.com) were applied to EAE data determine the effect size (difference in means) with 95% confidence interval (CI) for two or multi-group comparisons. Mann-Whitney test or t-test with Holm-Sidak correction for multiple comparisons was used for null hypothesis testing with alpha less than 0.05 considered significant; Fisher's exact test was used for two-group comparisons of categorical data; correlation was tested by linear regression and Pearson correlation testing (GraphPad Prism; San Diego, CA).

## Results

### Delayed axonal degeneration phenotype of *Sarm1* knockout neurons in culture

Prior research indicated that cultured neurons from *Sarm1* knockout mice show delayed axonal degeneration following axotomy or in response to oxidative stress [6,17]. To confirm the

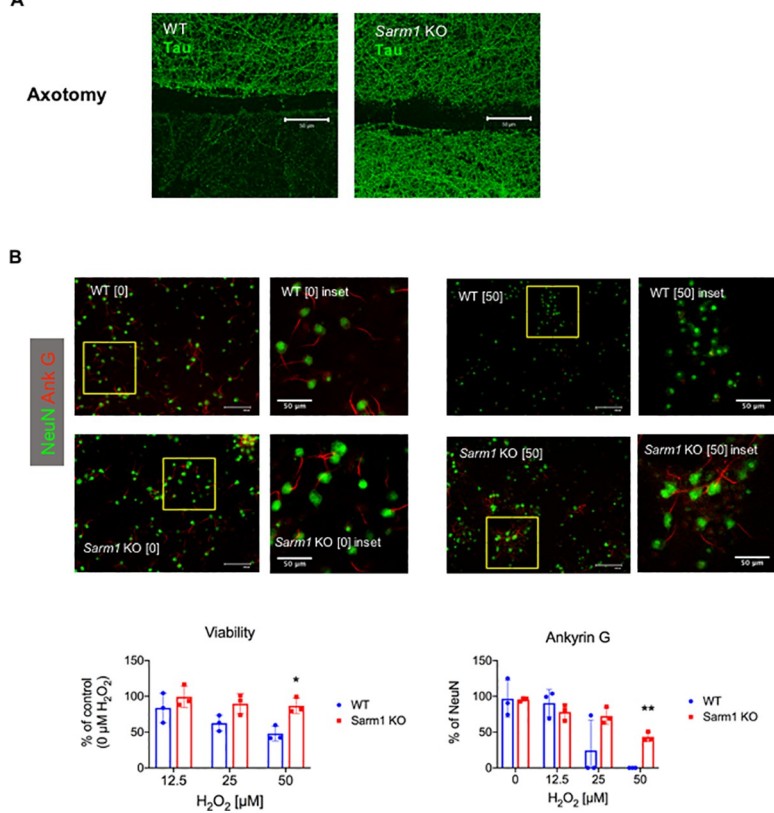

**Fig 1. Delayed axonal degeneration phenotype of *Sarm1* knockout (KO) neurons in culture.** (A) Cultured neurons from wild type (WT) and *Sarm1* KO mice were subjected to axotomy, then labeled for Tau 72 hours later. Confocal microscopy of Tau immunocytochemistry shows loss of Tau in distal axonal segments in WT, but not *Sarm1* KO, neurons at 72 h following axotomy. Representative images from 3 independent experiments. Scale bar, 50 μm. (B) Cultured neurons from WT and *Sarm1* KO mice were subjected to oxidative stress by 30 min exposure to 0, 12.5, 25 or 50 μM hydrogen peroxide ($H_2O_2$) then assayed 24 h later. Images show ankyrin G (red) and NeuN (green), detected by immunocytochemistry in cultured neurons from WT and *Sarm1* KO mice. Scale bar, 100 μm and inset scale bar 50 μm. Bar graph (left) shows mean neuronal viability +/- standard deviation at indicated $H_2O_2$ concentrations, assessed by dye exclusion and normalized to control (0 μM $H_2O_2$). * denotes p = 0.011, t-test with Holm-Sidak method for multiple comparisons. Bar graph (right) shows mean ankyrin G expression (% of NeuN) +/- standard deviation. ** denotes p < 0.001, t-test with Holm-Sidak method for multiple comparisons.

delayed axonal degeneration phenotype of *Sarm1* knockout mice, cultured neurons from *Sarm1* knockout and wild type mice were subjected to axotomy or exposed to hydrogen peroxide ($H_2O_2$). Tau immunocytochemistry 72 hours following axotomy of cultured neurons showed that whereas wild type axons lost Tau expression distal to axotomy, Tau expression was intact in distal axonal segments of *Sarm1* knockout neurons, indicating delayed axonal degeneration (Fig 1A). To test the neuroprotective effects of *Sarm1* deletion in the setting of oxidative stress, cortical neurons from *Sarm1* knockout and wild type mice were subjected to a brief (30 minute) exposure to $H_2O_2$ in culture then assayed 24 hours later for viability by dye exclusion and for ankyrin G expression as a marker of axonal integrity. Ankyrin G immunocytochemistry showed short linear proximal segment labeling consistent with axon initial segments (AIS) in *Sarm1* knockout and wild type neurons, with nearly 100% AIS expression on NeuN labeled cells in mock-treated (0 μM $H_2O_2$) neuronal cultures (Fig 1B). Following exposure to $H_2O_2$, neuronal viability was higher in *Sarm1* knockout neurons compared to wild type neurons (Fig 1B). Whereas exposure to $H_2O_2$ resulted in loss of AIS in wild type neurons,

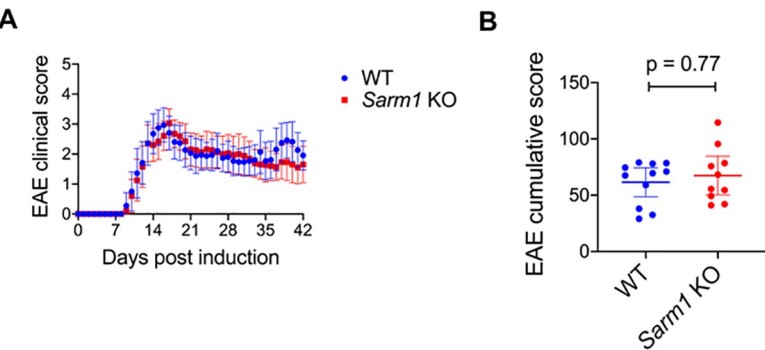

**Fig 2. Clinical course of EAE in *Sarm1* knockout (KO) mice compared to wild type (WT) littermates.** *Sarm1* KO mice and WT littermates were actively induced to undergo EAE and scored daily. (A) Mean daily clinical scores +/- 95% confidence interval. Data pooled from 8 independent experiments. N = 18 WT mice (11 females and 7 males) and N = 20 *Sarm1* KO mice (11 females and 9 males). (B) Mean cumulative clinical scores +/- 95% confidence interval for EAE-induced WT and *Sarm1* KO mice that were scored to 42 days post-induction. N = 11 WT mice (5 females and 6 males) and N = 10 *Sarm1* KO mice (5 females and 5 males). Mann-Whitney p-value.

*Sarm1* knockout neurons showed significant sparing of AIS compared to wild type at 50 μM $H_2O_2$ (Fig 1B). These results confirmed the delayed axonal degeneration and neuroprotective phenotype of *Sarm1* deletion, consistent with prior reports.

## Clinical course of EAE in *Sarm1* knockout mice compared to wild type littermates

To test the contribution of SARM1 to axonal degeneration in a model of chronic immune-mediated CNS inflammation, *Sarm1* knockout and wild type littermates were actively induced to undergo EAE. Incidence of EAE was similar between wild type and *Sarm1* knockout mice (100% vs. 96%; Fisher's exact test p > 0.99). Mortality associated with EAE was also similar between wild type and *Sarm1* knockout mice (17.4% vs. 18.5%; Fisher's exact test p > 0.99). Onset and peak severity of EAE were similar in *Sarm1* knockout mice and WT littermates (Fig 2A). Cumulative clinical scores did not differ substantially between *Sarm1* knockout and wild type littermates that were observed up to 42 days post-induction: mean difference of 5.97 [95% CI -10.4, 25.8; p = 0.77, Mann-Whitney test] (Fig 2B).

## Axonal degeneration is reduced early in the course of EAE in *Sarm1* knockout mice

To assess the effect of *Sarm1* deletion on axonal degeneration in EAE, *Sarm1* knockout mice and wild type littermates expressing neuronal YFP (i.e. *Sarm1-/-YFP+* and *Sarm1+/+YFP+*) were induced to undergo EAE. Control mice were injected with PBS in CFA and pertussis toxin. Neuronal YFP expression allowed identification of intact axons and fragmented axons that have undergone axonal transection and degeneration (Fig 3A). Analysis of YFP+ axons in the lumbar spinal cords of wild type EAE mice at 14 days post-induction showed evidence of axonal degeneration (Fig 3A and 3B). There was a reduction in total YFP+ axon counts in wild type, but not *Sarm1* knockout, EAE mice compared to wild type control mice: mean difference of -2.03 counts per field of view [95% CI -3.03, -0.86] for wild type EAE vs. wild type control; mean difference of 0.59 counts per field of view [95% CI -0.58, 2.54] for *Sarm1* knockout EAE vs. wild type control (Fig 3B). Fragmented YFP+ axon counts were significantly lower in *Sarm1* knockout EAE mice compared to wild type EAE mice at 14 days post-induction: mean difference of -1.43 counts per field of view [95% CI -1.85, -0.734; p = 0.029, Mann-Whitney]

**A**

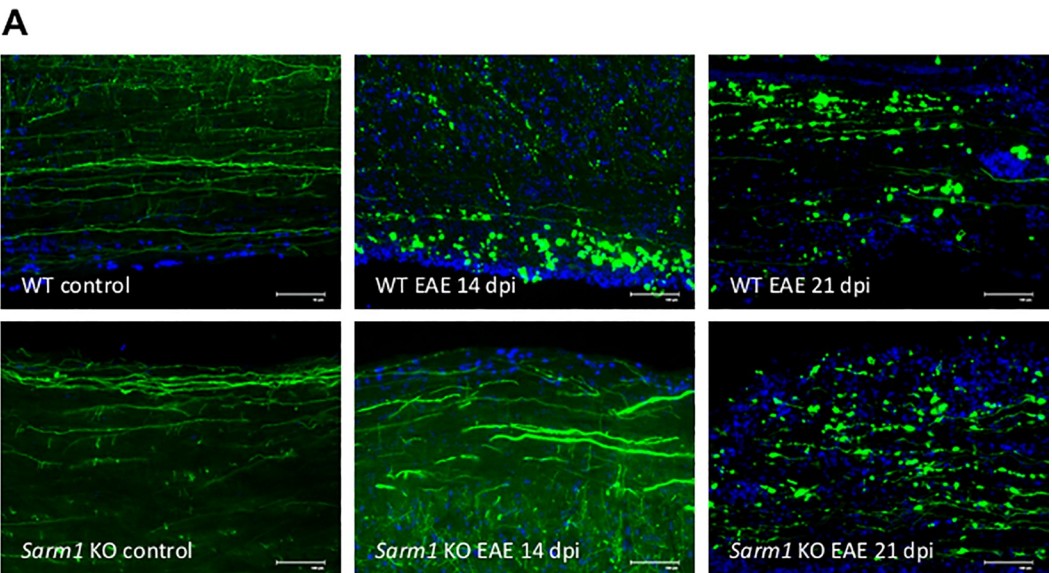

**B**  **YFP+ axon counts at 14 days post-induction**

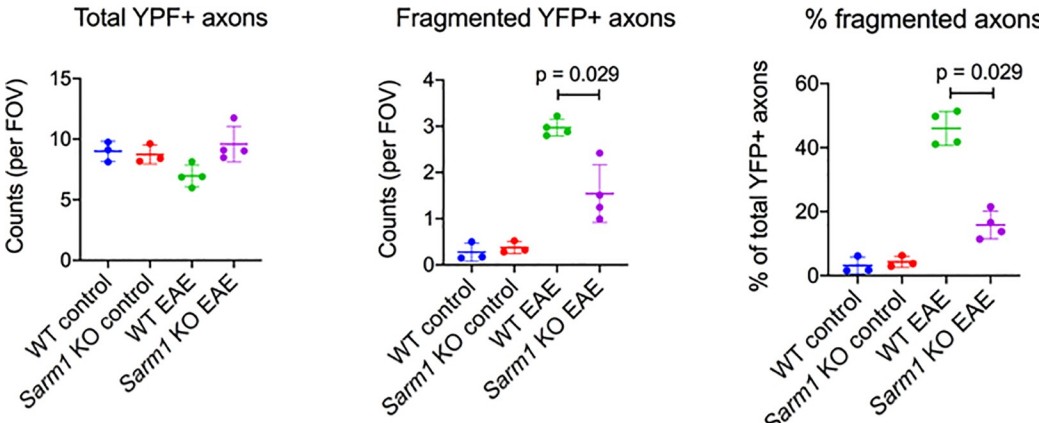

**C**  **YFP+ axon counts at 21 days post-induction**

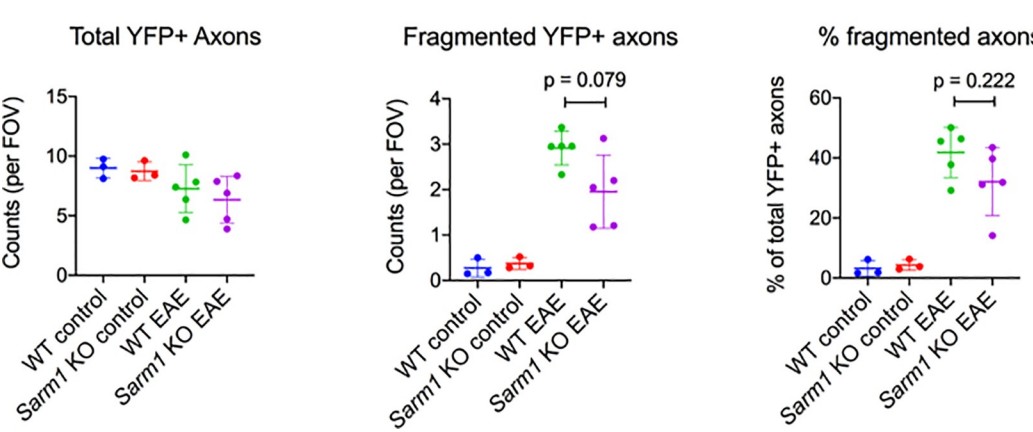

**Fig 3. Axonal degeneration is reduced early in the course of EAE in *Sarm1* knockout (KO) mice.** *Sarm1* KO mice and wild type (WT) littermates expressing neuronal yellow fluorescent protein (i.e. *Sarm1-/-*YFP+ and *Sarm1+/+*YFP+) were actively induced to undergo EAE (*Sarm1* KO EAE and WT EAE). Control mice were injected with phosphate buffered saline in complete Freund's adjuvant and pertussis toxin (*Sarm1* KO control and WT control). (A) Representative images showing YFP+ axons (green) in the lumbar cords of control and EAE mice at 14 and 21 days post-induction (dpi). DAPI nuclear stain (blue). Scale bar, 100 μm. (B) YFP+ axon counts at 14 dpi. Scatter plots show total (left) and fragmented (middle) YFP+ axon counts per field of view (FOV) for each animal by experimental groups. Right scatter plot shows percent (of total) fragmented YFP+ axons for each animal by experimental groups. Mean +/- standard deviation are indicated by line and error bars. N = 4 WT EAE (2 females and 2 males) and N = 4 *Sarm1* KO EAE (2 females and 2 males). Mann-Whitney p-value. (C) YFP+ axon counts at 21 dpi. Scatter plots show total (left) and fragmented (middle) YFP+ axon counts per FOV for each animal by experimental groups. Right scatter plot shows percent (of total) fragmented YFP+ axons for each animal by experimental groups. Mean +/- standard deviation are indicated by line and error bars. N = 5 WT EAE (4 females and 1 male) and N = 5 *Sarm1* KO EAE (3 females and 2 males). Mann-Whitney p-value.

(Fig 3A and 3B). Results were similar when fragmented YFP+ axon counts were analyzed as a percentage of total YFP+ axon counts. Percent fragmented YFP+ axons were significantly lower in *Sarm1* knockout EAE compared to wild type EAE at 14 days post-induction: mean difference of -30.2% [95% CI -35.7, -24.4; p = 0.029, Mann-Whitney] (Fig 3B). These results indicated that *Sarm1* knockout EAE mice showed less axonal degeneration at this early time point in the course of EAE. At 21 days post-induction, however, *Sarm1* knockout EAE mice also showed substantial burden of axonal degeneration (Fig 3A and 3C). There was a substantial reduction in total YFP+ axon counts in *Sarm1* knockout EAE mice compared to wild type control mice at 21 days post-induction: mean difference of -2.65 counts per field of view [95% CI -4.31, -0.934]. The difference in fragmented YFP+ axon counts between *Sarm1* knockout EAE and wild type EAE mice was smaller at 21 days then at 14 days post-induction, and did not reach statistical significance on null hypothesis testing: mean difference of -0.96 counts per field of view [95% CI -1.59, -0.176; p = 0.079, Mann-Whitney]. There was no significant difference between *Sarm1* knockout EAE mice and wild type EAE mice with respect to percent fragmented YFP+ axons at 21 days post-induction: mean difference of -9.72% [95% CI -21.7, 0.546; p = 0.22, Mann-Whitney] (Fig 3C). These results suggested that *Sarm1* deletion conferred neuroprotection early in the course of EAE, but that protection against axonal degeneration in *Sarm1* knockout mice was less robust later in the course of EAE.

## *Sarm1* knockout does not confer long-term protection against axonal degeneration in EAE

Axonal integrity was assessed at 42 days post-induction by immunohistochemistry to determine the effect of *Sarm1* deletion on long-term axonal degeneration in EAE. Accumulation of APP and non-phosphorylated neurofilament H (SMI-32) were used as markers of axonal injury [18,19]. Axonal density was assessed by SMI-31 immunohistochemistry to label phosphorylated neurofilament H [3]. Analysis of lumbar spine for APP immunohistochemistry showed no significant difference between *Sarm1* knockout and wild type littermates at 42 days post-induction of EAE: mean difference of 2.8 μm$^2$ per ROI [95% CI -11.0, 20.2; p = 0.841, Mann-Whitney] (Fig 4A). Neither was there a significant difference between *Sarm1* knockout mice and wild type littermates with respect to SMI-32 expressing axons at 42 days post-induction of EAE: mean difference of 66.0 μm$^2$ per ROI [95% CI -158, 378; p = 0.794, Mann-Whitney] (Fig 4B). Axonal density was assessed by SMI-31 immunohistochemistry (Fig 4C), and expressed as SMI-31+ area per hemicord. There was a significant inverse correlation between axonal density in the lumbar cords of EAE mice as measured by SMI-31 immunohistochemistry at 42 days post induction and their cumulative clinical scores: Pearson r = -0.606 [95% CI -0.836, -0.194; p = 0.0077] (Fig 4D), supporting the clinical relevance of this measure when assessed late in the course of EAE. Analysis of SMI-31 immunohistochemistry showed

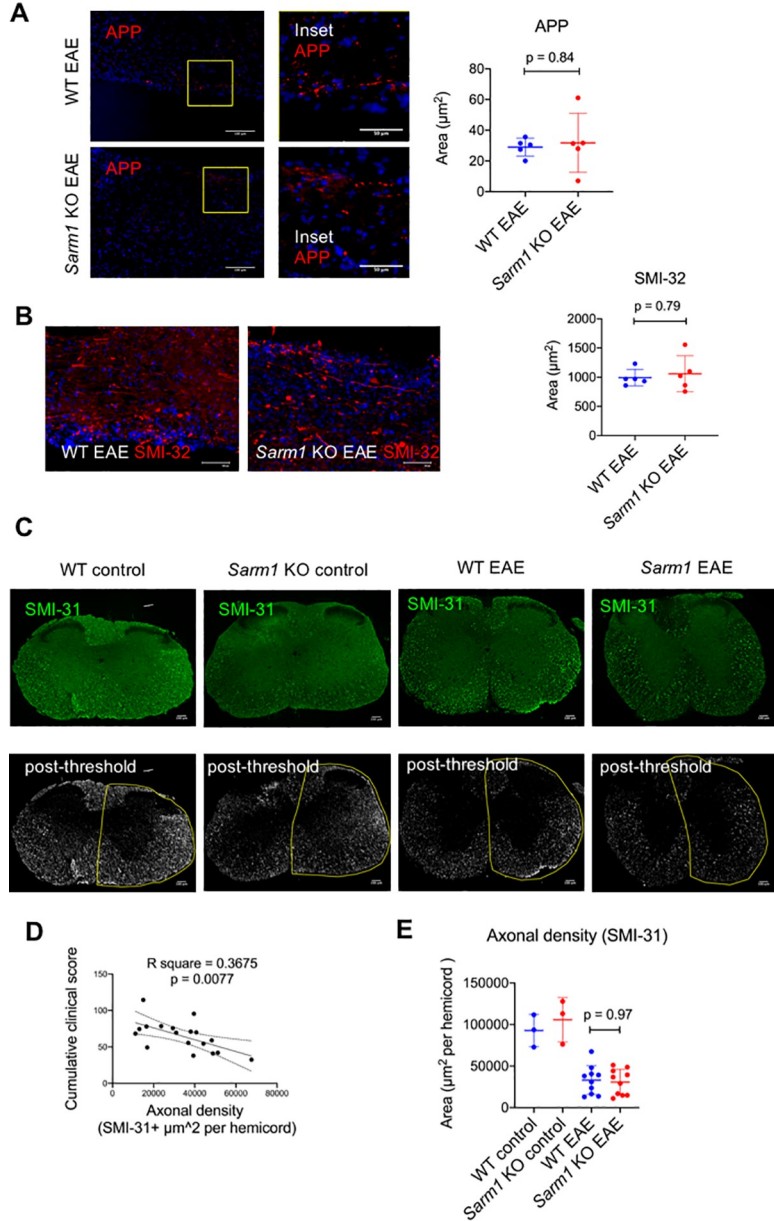

**Fig 4. *Sarm1* knockout does not confer long-term protection against axonal degeneration in EAE.** Lumbar cords from EAE-induced mice at 42 days post-induction were assessed by immunohistochemistry for amyloid precursor protein (APP) and non-phosphorylated neurofilament H (SMI-32) as markers of axonal injury and for phosphorylated neurofilament H (SMI-31) to assess axonal density. (A) Images show representative APP (red) immunohistochemistry on lumbar cord sections from EAE-induced *Sarm1* knockout mice (*Sarm1* KO EAE) and wild type littermates (WT EAE) showing APP-labeled axons and spheroids. DAPI nuclear stain (blue). Scale bar, 100 μm. Inset scale bar 50 μm. The scatter plot shows the mean APP+ area per region of interest for each animal. Line and error bars indicate group means and standard deviations. N = 5 WT EAE mice (4 females, 1 male) and N = 5 *Sarm1* KO EAE mice (3 females, 2 males). Mann-Whitney p-value. (B) Images shows representative SMI-32 (red) immunohistochemistry on lumbar cords from *Sarm1* KO EAE and WT EAE showing SMI-32 labeled axons and spheroids. DAPI nuclear stain (blue). Scale bar, 100 μm. The scatter plot shows the mean SMI-32+ area per region of interest for each animal. Line and error bars indicate means and standard deviations for each group. N = 5 WT EAE mice (4 females, 1 male) N = 5 *Sarm1* KO EAE mice (3 females, 2 males). Mann-Whitney p-value. (C) Images shows representative SMI-31 (green) immunohistochemistry on lumbar cords from *Sarm1* KO EAE and WT EAE mice. Control mice (WT control and *Sarm1* KO control) were immunized with phosphate buffered saline in complete Freund's adjuvant and pertussis toxin. Representative post-thresholding images with superimposed manual outline of hemicords are shown below to illustrate image analysis used to derive SMI-31+ area per hemicord. (D) Linear regression line with 95% confidence interval (dotted lines) showing the inverse relationship between axonal density (SMI-31+ area) and EAE cumulative

clinical score. (E) Scatter plot of mean axonal density per hemicord for each animal as measured by SMI-31 immunohistochemistry. Line and error bars indicate means and standard deviations for each group. N = 10 WT EAE mice (5 females, 5 males) and N = 10 *Sarm1* KO EAE mice (5 females, 5 males). Mann-Whitney p-value.

significant reduction in axonal density in the lumbar cords of both *Sarm1* knockout and wild type littermates at 42 days post-induction of EAE compared to control mice immunized with PBS in CFA and pertussis toxin: mean difference between wild type EAE and wild type control was $-5.96 \times 10^4$ μm$^2$ per hemicord [95% CI $-7.87 \times 10^4$, $-3.77 \times 10^4$], and mean difference between *Sarm1* knockout EAE and wild type control was $-6.20 \times 10^4$ μm$^2$ per hemicord [95% CI $-8.09 \times 10^4$, $-4.14 \times 10^4$] (Fig 4E). Two-group comparison between *Sarm1* knockout EAE and wild type EAE mice showed no significant difference in axonal density at 42 days post-induction: mean difference of $-2.39 \times 10^3$ μm$^2$ per hemicord [95% CI $-1.73 \times 10^4$, $1.02 \times 10^4$; $p = 0.971$, Mann-Whitney]. Together, these results indicated that *Sarm1* deletion did not result in long-term protection from axonal degeneration in EAE-induced mice.

## Peripheral immune activation and CNS immune cell infiltration are not altered in EAE-induced *Sarm1* knockout mice

Prior reports indicated that SARM1 inactivation could either promote or inhibit immune response depending on cell-type, disease model or species studied [11,15]. To exclude the possibility that SARM1 functions to influence the immune-mediated CNS inflammatory response of EAE and thereby alter its course, we compared the peripheral immune activation and CNS inflammatory cell infiltration associated with EAE in *Sarm1* knockout and wild type littermates. Splenocytes were obtained at 14 days post-induction of EAE and tested for *ex vivo* recall response to MOG$_{35-55}$ peptide, assaying T helper cytokine production and proliferation. There was no significant difference between *Sarm1* knockout and wild type EAE mice with respect to production of IL-4, IL-17 or IFN-γ (Table 1). In addition, there was no significant difference in the number of CD4+CD25+FOXP3+ regulatory T cells or in the number of proliferating T cells in response to ex vivo MOG$_{35-55}$ re-stimulation (Table 1). With respect to CNS

**Table 1. Peripheral immune activation and CNS immune cell infiltration.**

| Immune activation | N (WT EAE, Sarm1 KO EAE) | Effect size: *Sarm1* KO EAE vs. WT EAE [95% confidence interval] | Mann Whitney p |
|---|---|---|---|
| IL-4 (pg/ml) | 7, 10 | 3.03 [-4.74, 14.7] | 0.92 |
| IL-17 (pg/ml) | 7, 10 | 440 [−259, 1410] | 0.96 |
| IFN-γ (pg/ml) | 7, 10 | $5.31 \times 10^3$ [$-2.25 \times 10^3$, $1.22 \times 10^4$] | 0.30 |
| CD25+Foxp3+ (%CD3+CD4+) | 6, 7 | -1.14 [-5.86, 3.96] | 0.45 |
| CD4+Ki67+ (%CD3+) | 6, 7 | -0.751 [-2.34, 1.17] | 0.29 |
| CNS inflammatory cells (absolute counts) | N (WT EAE, Sarm1 KO EAE) | Effect size: *Sarm1* KO EAE vs. WT EAE [95% confidence interval] | Mann Whitney p |
| total leukocytes | 6, 7 | $-1.96 \times 10^5$ [$-5.26 \times 10^5$, $1.58 \times 10^5$] | 0.29 |
| Microglia | 6, 7 | $-2.42 \times 10^4$ [$-7.95 \times 10^4$, $2.47 \times 10^4$] | 0.53 |
| Macrophage | 6, 7 | $-1.13 \times 10^5$ [$-3.24 \times 10^5$, $9.51 \times 10^4$] | 0.37 |
| CD11b+Ly6Chi | 6, 7 | $-1.03 \times 10^5$ [$-2.39 \times 10^5$, $6 \times 10^4$] | 0.37 |
| CD11b+CD11c+ | 6, 7 | $-2.66 \times 10^4$ [$-9.88 \times 10^4$, $3.99 \times 10^4$] | 0.73 |
| T cells (CD3+) | 6, 7 | $-3.07 \times 10^4$ [$-8.96 \times 10^4$, $5.05 \times 10^4$] | 0.29 |
| CD3+CD4+ T cells | 6, 7 | $-8.15 \times 10^3$ [$-5.9 \times 10^4$, $4.73 \times 10^4$] | 0.95 |
| CD3+CD8+ T cells | 6, 7 | $-2.09 \times 10^4$ [$-6.62 \times 10^4$, $4.71 \times 10^2$] | 0.37 |
| B cells | 6, 7 | $2.31 \times 10^3$ [$-2.61 \times 10^3$, $9.32 \times 10^3$] | 0.73 |
| Neutrophils | 6, 7 | $-1.45 \times 10^4$ [$-4.76 \times 10^4$, $6.11 \times 10^3$] | 0.73 |

inflammatory cell infiltration, phenotypic enumeration of cells from thoracolumbar spines of EAE-induced mice by FACS at 14 days post-induction showed no significant difference between *Sarm1* knockout and wild type littermates with respect to the number of infiltrating immune cells (Table 1 and S1 Fig). These results suggest that SARM1 does not substantially influence peripheral immune activation or CNS inflammatory cell infiltration in EAE.

## Discussion

The contribution of SARM1-dependent program of axonal degeneration to the course and axonal pathology of immune-mediated CNS inflammatory disorders was previously unconfirmed. In this study, we compared the course, pathology and neuroimmunology of EAE in *Sarm1* knockout mice and wild type littermates to test whether or not the loss of SARM1 confers neuroprotection in an animal model of immune-mediated CNS inflammatory disorder.

We confirmed the delayed axonal degeneration phenotype of *Sarm1* knockout by demonstrating intact Tau protein expression in the distal stump of *Sarm1* knockout cultured neurons at 72 h following axotomy. In addition, cortical neurons from *Sarm1* knockout mice showed relative protection from cell death and axonal injury in the setting of oxidative stress, consistent with prior reports [6,17]. We assessed ankyrin G expression in the AIS as a marker of axonal integrity in cultured neurons. The loss of ankyrin G was previously shown to be a marker of axonal injury in EAE and in an *in vitro* model of oxidative stress-induced axonal degeneration [20,21]. Cultured neurons from *Sarm1* knockout showed relative preservation of ankyrin G expression in the AIS compared to wild type in the setting of oxidative stress. These *in vitro* results suggest a neuroprotective potential for SARM1 deletion in the setting of CNS inflammation, based on prior knowledge that oxidative stress is a key contributor to acute axonal injury during immune-mediated CNS inflammation [22,23].

The clinical course of EAE, however, did not differ substantially between *Sarm1* knockout and wild type littermates, suggesting that targeting SARM1 may not produce reliable clinical benefits in EAE or other immune-mediated CNS inflammatory disorders such as multiple sclerosis. There were no significant differences in the incidence, peak severity or mortality associated with EAE. At best, there was a modest difference in clinical scores at late time points. Effect size was small.

*Sarm1* deletion did not lead to long-term axonal protection, despite relative preservation of axonal integrity early in the course of EAE. Axonal degeneration of EAE was assessed using 2 approaches. Intact and fragmented YFP+ axons were assessed in *Sarm1* knockout and wild type littermates expressing neuronal YFP at 14 and 21 days post-induction of EAE [18]. Conventional immunohistochemistry was used to assess axonal integrity at 42 days post-induction. The results of YFP+ axon analysis suggested preservation of axonal integrity in *Sarm1* knockout mice at an early, but not late, time point in EAE. Whereas *Sarm1*+/+YFP+ mice showed substantial loss and fragmentation of YFP+ axons at 14 days post-induction, *Sarm1*-/-YFP+ mice were similar to control animals with respect to total YFP+ axon counts and showed significantly less fragmented YFP+ axon counts compared to *Sarm1*+/+YFP+ mice. However, at 21 days post-induction, *Sarm1*-/-YFP+ mice also showed substantial burden of axonal degeneration. Total YFP+ axon counts were lower in *Sarm1*-/-YFP+ mice compared to controls at 21 days post induction, and the degree of YFP+ axon fragmentation was not significantly lower compared to *Sarm1*+/+YFP+ mice at this time point. At 42 days post-induction, *Sarm1* knockout mice were indistinguishable from wild type littermates with respect to conventional immunohistochemical markers of axonal degeneration such as APP and SMI-32, and there was no significant difference between *Sarm1* knockout and wild type littermates in axonal density as measured by SMI-31 immunohistochemistry. There was a significant inverse

correlation between SMI-31+ axonal density at 42 days post induction and cumulative clinical scores, supporting the clinical relevance of axonal loss to neurological dysfunction late in the course of EAE. Together, these results indicated that *Sarm1* deletion conferred early, but not long-term, axonal protection in EAE.

The key question addressed in this study was to what extent does programmed axonal degeneration contribute to neurodegeneration in immune-mediated CNS inflammatory disorders. Two proteins involved in NAD metabolism, SARM1 and NMNAT, are key opposing elements of a mechanism of programmed axonal degeneration [5–7]. The loss of NMNAT or the activation of SARM1 triggers intra-axonal depletion of NAD, leading to local energetic failure culminating in axonal degradation [24]. Prior research on the contribution of NMNAT to axonal degeneration in CNS inflammatory disorders produced conflicting results. Using the slow Wallerian degeneration (*Wld^s*) mice, which express a chimeric form of the axon survival factor NMNAT, one study reported a modest reduction in long-term axonal loss in the *Wld^s* mice undergoing EAE [25]. However, a second study showed no difference in long-term axonal loss in the EAE-induced *Wld^s* mice [4]. Mechanistically, SARM1 is downstream of NMNAT, and SARM1 inactivation can rescue the axon from degeneration even after the loss of NMNAT [26]. Prior research also showed that *Sarm1* deletion conferred substantially longer neuroprotection over *Wld^s* in an animal model of axonopathy [27]. Therefore, it remained plausible that *Sarm1* knockout mice might be more informative than the *Wld^s* mice with respect to delineating the full contribution of programmed axonal degeneration to the pathophysiology of EAE. The results of the current study, however, indicate that targeting SARM1 is no more likely than NMNAT to reduce long-term axonal degeneration associated with EAE. Considered together with results from prior research, the results presented herein suggest that SARM1/NMNAT-dependent mechanism of programmed axonal degeneration makes a modest contribution, if at all, to the overall burden of long-term axonal degeneration in this model of immune-mediated CNS inflammatory disorder.

Targeting SARM1 is neuroprotective in some models of neurological injury, but not others. *Sarm1* knockout mice showed reduced clinical severity and reduced axonal loss in animal models of traumatic brain injury [10,28]. *Sarm1* deletion was beneficial in the TDP-43$^{Q331K}$ model of amyotrophic lateral sclerosis-frontotemporal dementia [29], but had no significant impact on axonal loss or clinical course of mutant SOD1 model of amyotrophic lateral sclerosis [30]. Therefore, the contribution of SARM1 is likely to be disease-specific. A number of interdependent pathogenic mechanisms have been proposed to explain inflammation-induced axonal degeneration in multiple sclerosis and EAE. These include oxidative stress [22], mitochondrial dysfunction [31], intra-axonal ionic dyshomeostasis [32] and excitotoxicity [33]. Prior research suggests that a distinction should be made between acute and chronic axonal injury in EAE and multiple sclerosis [34]. One possible explanation for the finding that *Sarm1* deletion results in early but not late axonal protection is that the primary mechanistic drivers of axonal degeneration may differ between early and late axonal degeneration in EAE and multiple sclerosis. Whereas oxidative stress is thought to play a prominent role in acute axonal degeneration, which can occur independently of demyelination [23], intra-axonal ionic dyshomeostasis may be more critical to axonal degeneration along chronically demyelinated axons [35]. *Sarm1* deletion may be more neuroprotective in the setting of oxidative stress associated with early, acute inflammation in EAE, but not as protective against axonal degeneration that occurs as a result of chronic demyelination. Alternatively, the results of the study may indicate that *Sarm1* deletion simply delays the perhaps inevitable axonal degeneration that follows axonal transection, similar to that seen following axotomy [6].

The immune response and CNS inflammation in *Sarm1* knockout mice were also examined in this study. Prior research showed that SARM1 function in immune cells can affect CNS

inflammation [14,15]. Significant alteration in the immune-mediated CNS inflammatory response of EAE in *Sarm1* knockout mice could have confounded the interpretation of the results, potentially masking a neuroprotective effect. The results of this study indicate that *Sarm1* deletion does not lead to a significant alteration in peripheral immune activation or CNS inflammatory cell infiltration in the context of EAE. MOG-specific immune activation and CNS inflammatory infiltration were similar in *Sarm1* knockout and wild type littermates. The lack of a significant change immune activation was further supported by the clinical data that showed similar onset and incidence of EAE between *Sarm1* knockout mice and wild type littermate. These results are also consistent with a recent report that indicates that background effects in part explain previously reported effects of *Sarm1* deletion on immune response, suggesting a more limited role for SARM1 in immunity [36].

This study has several limitations. Axonal YFP epifluorescence in the Thy1-YFP-H transgenic mice is susceptible to fluorescence quenching associated with blood-brain barrier breakdown and local edema [37]. Although YFP fluorescence quenching could lead to spuriously low total YFP+ axon counts, it is unlikely to result in the increased fragmented YFP+ axon counts that were observed in wild type and *Sarm1* knockout EAE mice at 14 and 21 days post-induction, respectively. Immune responses were studied at day 14 post-induction, but not at other time points. The possibility that *Sarm1* deletion might affect CNS inflammation earlier or later in the course of EAE has not been excluded. A broad, but not exhaustive, survey of CNS infiltrating immune cell subsets was performed in this study. The effect of *Sarm1* deletion on demyelination was not examined in this study. SARM1 function is not known to be involved in myelin formation or oligodendrocyte survival, and thus *Sarm1* deletion is unlikely to directly affect myelination. However, we have not excluded the possibility that early axonal protection may indirectly affect remyelination in EAE. Further work is needed to ascertain the impact of *Sarm1* deletion on demyelination/remyelination in the setting of CNS inflammation. Another limitation of the study is that EAE recapitulates many but not all aspects of axonal degeneration observed in multiple sclerosis [38]. EAE alone may not adequately capture the impact of targeting SARM1 on neurodegeneration in multiple sclerosis.

In conclusion, *Sarm1* deletion conferred protection from axonal degeneration early in the course of EAE, but did not confer long-term protection against axonal degeneration in this model of immune-mediated CNS inflammation.

## Supporting information

**S1 Fig. Phenotypic enumeration of CNS inflammatory cells by FACS analysis.** Representative plots of FACS analysis to enumerate CNS inflammatory cells from thoracolumbar cords of EAE mice are shown. A) Gating for singlets B) Gating for Count beads and live cells. C) Total CD45+ (leukocyte) gating. D) CD45+CD3+ (T cell) gating. E) CD45+Ly6G+ (neutrophil) gating. F) Ly6G-CD11b+CD11c+ (CD11b+ dendritic cell) gating. G) Ly6G-CD11b+Ly6C$^{hi}$ (Ly6C high monocyte/macrophage) gating. H) Ly6G-CD11b+CD45$^{int}$ (microglia) and Ly6G-CD11b+CD45$^{hi}$ (macrophage) gating. I) CD3+CD4+ (CD4 T cell) gating. J) CD3+CD8+ (CD8 T cell) gating. K) CD19+ (B-cell) gating.
(TIF)

## Acknowledgments

Microscopy and flow cytometry were performed at the VCU Department of Anatomy & Neurobiology Microscopy Facility and at the VCU Massey Cancer Center Flow Cytometry Shared Resource.

## Author Contributions

**Conceptualization:** Kenneth Viar, Unsong Oh.

**Formal analysis:** Kenneth Viar, Daniel Njoku, Julie Secor McVoy, Unsong Oh.

**Investigation:** Kenneth Viar, Daniel Njoku, Julie Secor McVoy, Unsong Oh.

**Writing – original draft:** Kenneth Viar, Unsong Oh.

**Writing – review & editing:** Daniel Njoku, Julie Secor McVoy, Unsong Oh.

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
