## [Decision Letter · Decision Letter 0]

6 May 2020

PONE-D-20-10362

Sarm1 knockout delays but does not protect against axonal degeneration in experimental allergic encephalomyelitis

PLOS ONE

Dear Dr. Oh,

Thank you for submitting your manuscript to PLOS ONE. After careful consideration, we feel that it has merit but does not fully meet PLOS ONE’s publication criteria as it currently stands. Therefore, we invite you to submit a revised version of the manuscript that addresses the points raised during the review process.

Moreover, attention must be paid to not overstating the conclusions of the manuscript and acknowledging the limitations of this model.

We would appreciate receiving your revised manuscript by Jun 20 2020 11:59PM. To enhance the reproducibility of your results, we recommend that if applicable you deposit your laboratory protocols in protocols.io, where a protocol can be assigned its own identifier (DOI) such that it can be cited independently in the future. For instructions see: http://journals.plos.org/plosone/s/submission-guidelines#loc-laboratory-protocols

We look forward to receiving your revised manuscript.

Kind regards,

Thomas Forsthuber

Academic Editor

PLOS ONE

Journal Requirements:

Reviewers' comments:

Reviewer's Responses to Questions

**Comments to the Author**

1. Is the manuscript technically sound, and do the data support the conclusions?

Reviewer #1: Yes

Reviewer #2: No

2. Has the statistical analysis been performed appropriately and rigorously? 

Reviewer #1: Yes

Reviewer #2: Yes

3. Have the authors made all data underlying the findings in their manuscript fully available?

Reviewer #1: No

Reviewer #2: Yes

4. Is the manuscript presented in an intelligible fashion and written in standard English?

Reviewer #1: Yes

Reviewer #2: Yes

5. Review Comments to the Author

Reviewer #1: This is a clear and well designed study. The findings were communicated well and will be useful to the research community. Overall I had only relatively minor concerns:

1. The phenotypic data currently available in the supporting information is difficult to follow. A more extensive figure legend would be useful to be certain that the reader understands the data presented, including what populations are shown in each section and what gates are applicable to individual graphs.

2. The authors conclude that " SARM1 activation is not the critical trigger for irreversible axonal injury in CNS inflammatory disorders" and suggest that SARM1 involvement in axonal loss is insult dependent. However the authors do not note that this intermediate outcome, protection from early loss with slightly later loss of axon integrity is not typical of previous reports regarding SARM1 activity. The finding that SARM1 appears to block axonal damage at the peak of disease could, and should be discussed in greater detail. This will probably also have to modify the authors statement that "There was a significant inverse correlation between SMI-31+ axonal

density and cumulative clinical scores, supporting the clinical relevance of axonal loss to neurological

dysfunction in EAE."

3. The authors note that the lack of examination of demyelination is a limitation. It is worth discussing how the results reported may impact myelination/demyelination.

Reviewer #2: The authors test the EAE model of win wild type and SARM1 KO mice. SARM1 is a key driver of pathological axon degeneration, and so the authors assess whether SARM1 blocks axon loss in the EAE model of neuroinflammation. They find that SARM1 KO has a significant delay in axon loss, but that axon loss is not prevented indefinitely. They do not see any difference in measures of immune activation of disease score between wild type and SARM1 KO mice. They conclude that SARM1 KO delays but does not protect against axon degeneration in EAE and that SARM1 "is not the critical trigger for irreversible axonal injury in CNS inflammatory disorders." While this study is novel and performed is technically of high quality, some of the conclusions are not well supported by the data.

1) The authors show that axons are well protected for about 2 weeks, and then degenerate in the SARM1 Ko during EAE. This is almost the identical time course to the protection afforded by SARM1 following axotomy in vivo. Axons survive for about 2 weeks and then degenerate in a SARM1-independent manner, likely due to the absence of cell-body derived factors. In EAE, it is very likely that the immune attack causes severing of axons and so it quite similar to axotomy. Do the authors see evidence for axon severing in their wild type mice? This would also explain the inability of SARM1 KO to change the disease course. Severed axons obviously can't contribute to function. The authors should acknowledge that all their data can be explained by the occurrence of axon severing in the EAE model.

2) Based on their EAE results, the authors making sweeping conclusions about the role of SARM1 in CNS neuroinflammatory disease. This is dramatically overstating what can be concluded from this model. While it is clear that the EAE model is a reasonable model of the inflammatory component of MS, it is not known if EAE models the neurodegenerative component of MS. Indeed, based on the time course it seems quite unlikely. Hence, it is inappropriate to conclude that SARM1 is not involved in CNS neuroinflammatory disorders. These statements should be removed, and the limitations of the EAE model for studying the neurodegenerative component of MS should be clearly acknowledged.

3) The title states that SARM1 KO does not "protect against axonal degeneration" in EAE. However the data clearly show that it does "protect," even if it doesn't permanently block. Again, this is identical to the result with axotomy--nothing can permanently block degeneration of a severed axon. The title should be changed to reflect that SARM1 Ko does protect against axon degeneration in EAE, albeit not indefinitely.

6. PLOS authors have the option to publish the peer review history of their article (what does this mean?). If published, this will include your full peer review and any attached files.

Reviewer #1: No

Reviewer #2: No

---

## [Author Response · Author response to Decision Letter 0]

12 May 2020

Response to PLOS ONE reviewer comments

PONE-D-20-10362

Dear PLOS ONE Reviewers

Thank you for your constructive comments. The manuscript has been revised in response to your comments.

Reviewer #1

1. The phenotypic data currently available in the supporting information is difficult to follow. A more extensive figure legend would be useful to be certain that the reader understands the data presented, including what populations are shown in each section and what gates are applicable to individual graphs

A more extensive figure legend has been provided for Supporting information S1. Fig 1.

2. The authors conclude that " SARM1 activation is not the critical trigger for irreversible axonal injury in CNS inflammatory disorders" and suggest that SARM1 involvement in axonal loss is insult dependent. However the authors do not note that this intermediate outcome, protection from early loss with slightly later loss of axon integrity is not typical of previous reports regarding SARM1 activity. The finding that SARM1 appears to block axonal damage at the peak of disease could, and should be discussed in greater detail. This will probably also have to modify the authors statement that "There was a significant inverse correlation between SMI-31+ axonal density and cumulative clinical scores, supporting the clinical relevance of axonal loss to neurological dysfunction in EAE."

The statement “SARM1 activation is not the critical trigger for irreversible axonal injury in CNS inflammatory disorders” has been removed in the revised manuscript. An alternative explanation for the finding that Sarm1 deletion protects early but not late axonal degeneration has been proposed in the discussion, based on mechanistic distinction between acute and chronic axonal degeneration in EAE and multiple sclerosis. The manuscript was also revised to qualify the statement regarding the inverse correlation between SMI-31+ axonal density and cumulative clinical scores to specify that this correlation applies to a late time point in the course of EAE. 

3. The authors note that the lack of examination of demyelination is a limitation. It is worth 

discussing how the results reported may impact myelination/demyelination. 

The manuscript has been revised to discuss possible impact on myelination/demyelination: “SARM1 function is not known to be involved in myelin formation or oligodendrocyte survival, and thus Sarm1 deletion is unlikely to directly affect myelination. However, we have not excluded the possibility that early axonal protection may indirectly affect remyelination in EAE. Further work is needed to ascertain the impact of Sarm1 deletion on demyelination/ remyelination in the setting of CNS inflammation.”

Reviewer #2

1) The authors show that axons are well protected for about 2 weeks, and then degenerate in the SARM1 Ko during EAE. This is almost the identical time course to the protection afforded by SARM1 following axotomy in vivo. Axons survive for about 2 weeks and then degenerate in a SARM1- independent manner, likely due to the absence of cell-body derived factors. In EAE, it is very likely that the immune attack causes severing of axons and so it quite similar to axotomy. Do the authors see evidence for axon severing in their wild type mice? This would also explain the inability of SARM1 KO to change the disease course. Severed axons obviously can't contribute to function. The authors should acknowledge that all their data can be explained by the occurrence of axon severing in the EAE model. 

The manuscript has been revised to acknowledge the consequence of axonal transection: “Alternatively, the results of the study may indicate that Sarm1 deletion simply delays the perhaps inevitable axonal degeneration that follows axonal transection, similar to that seen following axotomy”

2) Based on their EAE results, the authors making sweeping conclusions about the role of SARM1 in CNS neuroinflammatory disease. This is dramatically overstating what can be concluded from this model. While it is clear that the EAE model is a reasonable model of the inflammatory component of MS, it is not known if EAE models the neurodegenerative component of MS. Indeed, based on the time course it seems quite unlikely. Hence, it is inappropriate to conclude that SARM1 is not involved in CNS neuroinflammatory disorders. These statements should be removed, and the limitations of the EAE model for studying the neurodegenerative component of MS should be clearly acknowledged. 

The manuscript has been revised to remove the statement “SARM1 activation is not the critical trigger for irreversible axonal injury in CNS inflammatory disorders”. An alternative explanation has been proposed in the discussion based on mechanistic distinction between acute and chronic axonal degeneration in EAE and multiple sclerosis. Furthermore, the manuscript has been revised to acknowledge the limitations of EAE in modeling neurodegeneration of multiple sclerosis.

3) The title states that SARM1 KO does not "protect against axonal degeneration" in EAE. However the data clearly show that it does "protect," even if it doesn't permanently block. Again, this is identical to the result with axotomy--nothing can permanently block degeneration of a severed axon. The title should be changed to reflect that SARM1 Ko does protect against axon degeneration in EAE, albeit not indefinitely. 

In response to reviewer comments, the manuscript title has been revised to “Sarm1 knockout protects against early but not late axonal degeneration in experimental allergic encephalomyelitis”. In addition the conclusion has also been revised to state early but not late protection of axonal degeneration in Sarm1 KO in EAE.

---

## [Decision Letter · Decision Letter 1]

2 Jun 2020

PONE-D-20-10362R1

Sarm1 knockout protects against early but not late axonal degeneration in experimental allergic encephalomyelitis

PLOS ONE

Dear Dr. Oh,

Thank you for submitting your manuscript to PLOS ONE. After careful consideration, we feel that it has merit but does not fully meet PLOS ONE’s publication criteria as it currently stands. Therefore, we invite you to submit a revised version of the manuscript that addresses the points raised during the review process.

Please make sure to address the remaining minor concerns.

We look forward to receiving your revised manuscript.

Kind regards,

Thomas Forsthuber

Academic Editor

PLOS ONE

Reviewers' comments:

Reviewer's Responses to Questions

**Comments to the Author**

1. If the authors have adequately addressed your comments raised in a previous round of review and you feel that this manuscript is now acceptable for publication, you may indicate that here to bypass the “Comments to the Author” section, enter your conflict of interest statement in the “Confidential to Editor” section, and submit your "Accept" recommendation.

Reviewer #1: All comments have been addressed

Reviewer #2: (No Response)

2. Is the manuscript technically sound, and do the data support the conclusions?

Reviewer #1: Yes

Reviewer #2: Yes

3. Has the statistical analysis been performed appropriately and rigorously? 

Reviewer #1: Yes

Reviewer #2: Yes

4. Have the authors made all data underlying the findings in their manuscript fully available?

Reviewer #1: Yes

Reviewer #2: Yes

5. Is the manuscript presented in an intelligible fashion and written in standard English?

Reviewer #1: Yes

Reviewer #2: Yes

6. Review Comments to the Author

Reviewer #1: (No Response)

Reviewer #2: The authors made most of the requested changes but missed a couple of spots. In addition there are a few missing references.

1) I had asked the authors not claim that there findings were valid for all CNS inflammatory disorders, but just for this one model. They made the changes most places, but missed line 451 and line 471. Please change to "model of" here as well.

2) The authors discuss the potential role of SARM1 in the immune system, but do not mention an important new paper that suggests most of the referenced studies are an artifact of the genetic background in the line used by the authors. The authors should acknowledge this issue and reference the paper (Uccellini 2020).

3) The authors mention one study that did not fine a role for SARM1 in an ALS model, but leave out another study that did find a role (White 2019). In fact this paper was similar to the author's work, showing some axonal protection but not behavioral protection. This second study should be acknowledged.

7. PLOS authors have the option to publish the peer review history of their article (what does this mean?). If published, this will include your full peer review and any attached files.

Reviewer #1: No

Reviewer #2: No

---

## [Author Response · Author response to Decision Letter 1]

5 Jun 2020

PONE-D-20-10362R1

Response to Reviewers

Reviewer #2: The authors made most of the requested changes but missed a couple of spots. In addition there are a few missing references. 1) I had asked the authors not claim that there findings were valid for all CNS inflammatory disorders, but just for this one model. They made the changes most places, but missed line 451 and line 471. Please change to "model of" here as well. 2) The authors discuss the potential role of SARM1 in the immune system, but do not mention an important new paper that suggests most of the referenced studies are an artifact of the genetic background in the line used by the authors. The authors should acknowledge this issue and reference the paper (Uccellini 2020). 3) The authors mention one study that did not fine a role for SARM1 in an ALS model, but leave out another study that did find a role (White 2019). In fact this paper was similar to the author's work, showing some axonal protection but not behavioral protection. This second study should be acknowledged. 

The manuscript has been revised in response to reviewer comments. 

1) We agree with the reviewer and line 471 has been changed in the revised manuscript to:

471 “... degeneration in this model of immune-mediated CNS inflammatory disorder.”

However, we are confounded by the reviewer’s request for change in line 451 which reads as follows in the resubmitted manuscript: 

451 “Sarm1 deletion conferred early, but not long-term, axonal protection in EAE.”

There is no claim to validity for CNS inflammatory disorders in general in line 451, therefore we are at a loss as to how that should to be changed.

2) The paper by Uccellini et al reporting that background effects in part explain some of the previously reported effect of SARM1 on immune response has been acknowledged and referenced in the revised Discussion. 

3) The paper by White et al was also acknowledged in the revised Discussion.

---

## [Decision Letter · Decision Letter 2]

10 Jun 2020

Sarm1 knockout protects against early but not late axonal degeneration in experimental allergic encephalomyelitis

PONE-D-20-10362R2

Dear Dr. Oh,

We’re pleased to inform you that your manuscript has been judged scientifically suitable for publication and will be formally accepted for publication once it meets all outstanding technical requirements.

Kind regards,

Thomas Forsthuber

Academic Editor

PLOS ONE

Additional Editor Comments (optional):

Reviewers' comments:

Reviewer's Responses to Questions

**Comments to the Author**

1. If the authors have adequately addressed your comments raised in a previous round of review and you feel that this manuscript is now acceptable for publication, you may indicate that here to bypass the “Comments to the Author” section, enter your conflict of interest statement in the “Confidential to Editor” section, and submit your "Accept" recommendation.

Reviewer #2: All comments have been addressed

2. Is the manuscript technically sound, and do the data support the conclusions?

Reviewer #2: Yes

3. Has the statistical analysis been performed appropriately and rigorously? 

Reviewer #2: Yes

4. Have the authors made all data underlying the findings in their manuscript fully available?

Reviewer #2: Yes

5. Is the manuscript presented in an intelligible fashion and written in standard English?

Reviewer #2: Yes

6. Review Comments to the Author

Reviewer #2: (No Response)

7. PLOS authors have the option to publish the peer review history of their article (what does this mean?). If published, this will include your full peer review and any attached files.

Reviewer #2: No

---

## [Editor Report · Acceptance letter]

15 Jun 2020

PONE-D-20-10362R2 

Sarm1 knockout protects against early but not late axonal degeneration in experimental allergic encephalomyelitis 

Dear Dr. Oh:

I'm pleased to inform you that your manuscript has been deemed suitable for publication in PLOS ONE. Congratulations! Your manuscript is now with our production department. 

Kind regards, 

on behalf of

Dr. Thomas Forsthuber 

Academic Editor

PLOS ONE